# The Impact of Decarbonization Scenarios on Air Quality and Human Health in Poland—Analysis of Scenarios up to 2050

**Janusz Zyśk** *[ID], **Artur Wyrwa**[ID], **Wojciech Suwała, Marcin Pluta, Tadeusz Olkuski** and **Maciej Raczyński**[ID]

Faculty of Energy and Fuels, AGH University of Science and Technology, 30-059 Kraków, Poland; awyrwa@agh.edu.pl (A.W.); suwalaw@agh.edu.pl (W.S.); mpluta@agh.edu.pl (M.P.); olkuski@agh.edu.pl (T.O.); makracz@agh.edu.pl (M.R.)
* Correspondence: jazysk@agh.edu.pl; Tel.: +48-12-617-41-74

**Abstract:** Poland faces two great challenges in the field of environment and atmosphere protection: improving air quality, especially by reducing particulate matter (PM) emissions, and reducing relatively high greenhouse gas emissions. The aim of this research was to investigate how the fuel and technological transformations in the power, road transport, and household and tertiary sectors aimed at reducing carbon dioxide ($CO_2$) emissions in Poland would affect air quality, human health, and the associated external costs. The study was conducted for 2050 while considering 2015 as the base year. Ambient PM2.5 (particles with a diameter of less than 2.5 μm) concentration was used as a proxy air quality indicator. The analysis was based on decarbonization scenarios developed within the REFLEX Project (Analysis of the European energy system under the aspects of flexibility and technological progress). The three scenarios of the REFLEX Project focused on the reduction of $CO_2$ emissions up to 2050 from various sectors, mainly by the means of fuel and technological switches. This also led to the changes in the emission levels of pollutants that directly affect air quality, which were calculated with the use of fuel- and technology-specific emission factors. Next, for each emission scenario, ambient concentrations of PM2.5 and others pollutants were calculated with the use of the Polyphemus—an Eulerian-type air quality modelling system. Subsequently, the health impact of population exposed to air pollution and associated external costs were calculated using the πESA (Platform for Integrated Energy System Analysis) platform. The health impacts considered were the number of years of life lost, restricted activity days, and number of chronic bronchitis cases. The results showed that the largest reductions in both greenhouse gas and PM emissions—and consequently improvements of air quality resulting in a decrease of negative impacts on human health and a decrease of external costs—can be achieved by the transformation of heat production in the household and tertiary sector. The results also showed that the decrease in PM2.5 emissions envisaged in the analyzed scenarios in 2050 will lead to a reduction in the number of lost years of life by about 35 thousand and an avoidance of external costs by EUR 2.4 billion.

**Keywords:** external cost; air quality modeling; emission scenarios; low-stack emission; $CO_2$; PM2.5

## 1. Introduction

Poland has some the most polluted air in all of the European Union, particularly facing a significant problem with a particulate matter and benzo(a)pyrene concentrations. The limit values set for the ambient concentration of pollutants considered in the directives on ambient air quality and cleaner air for Europe (CAFE) are often exceeded [1–3]. These pollutants are mainly emitted from low-stack

sources (below 40 m above the ground), most of them from the household and transport sectors [3,4]. The pollutants emitted close to the ground level are relatively poorly transported in the atmosphere, which means that they have a significant impact on local air quality and human health. In Poland, most houses are heated with coal in low-efficiency stoves and boilers. The shares of household and tertiary, power sector, and road transport in PM2.5 (particulate matter with particles with a diameter of less than 2.5 μm) emissions in Poland in 2017 were 38%, 5%, and 9%, respectively [4]. These sectors also had similar shares in the case of PM10 (particles with a diameter of less than 10 μm) emissions. At present, almost 80% of carcinogenic benzo(a)pyrene is emitted from the household and tertiary sector. The World Health Organization's (WHO) statistics show that the mortality rate due to poor air quality in Poland was 36.3 (36.3 deaths per 100,000 inhabitants) in 2018 [5]. In an analysis by the European Environmental Agency (EEA) in 2018, the number of the year of life lost for each 100,000 inhabitants was estimated at 1364 for PM2.5, 49 for nitrogen dioxide ($NO_2$), and 36 for ozone ($O_3$) [1]. In comparison, the relevant average values for EU-28 were 800 in case of PM2.5, 100 for $NO_2$, and 30 for $O_3$. The studies of the EEA were based on the same approach as that proposed in this work, which considers the concentration of pollutants, population density, and concentration–response functions.

Particulate matter, the PM2.5 fraction in particular, is considered to be harmful to human health. Due to its small size, PM2.5 are able to pass through the barrier of the upper (nose and pharynx) and lower (larynx, trachea) respiratory tracts and finally enter the lungs, from where they can be distributed by the circulatory system throughout the human body [6,7]. PM2.5 have been found to significantly increase the occurrence of many diseases including asthma, allergies, chronic obstructive pulmonary disease, heart failure, carcinoma, and stroke [6–9]. These various effects can be expressed with the use of standardized indicators (mortality rate and years of life lost) to which a monetary value can be assigned to represent the associated health costs (e.g., lost wages and hospital admission expenditures). This makes it possible to consider environmental and health impacts in the decision making process while developing policies and action plans, as well as comparing alternative development pathways for regions and economic sectors [8–12].

Apart from poor local air quality, Poland is characterized by relatively high $CO_2$ emissions. In Poland, almost 80% of electricity is generated by burning hard coal and lignite [13]. Electricity production using natural gas makes up 8%. The share of energy from renewable sources (RES) in final energy consumption in Poland in 2018 was 11.6% [14]. The goal for 2020 in Poland regarding RES was 15% [15]. The total $CO_2$ emission in Poland in 2017 was 299.1 million tons. The public electricity and heat production sector emitted 155.5 million tons, road transport emitted 62.5 million tons, and the household and tertiary sector emitted 43 million tons. These three sectors remain the main sources of $CO_2$ and have the greatest potential to reduce carbon emissions [16].

According to the EU 2016/2284 directive, all member states are obliged to reduce their emissions in accordance with their individual country targets [17]. In Poland, by 2020 (compared to 2005) it was decided to be necessary to reduce sulfur dioxide ($SO_2$) emissions by 59%, nitrogen oxide ($NO_x$) by 30%, non-methane volatile organic compounds (NMVOC) by 25%, ammonia ($NH_3$) by 1%, and PM2.5 by 16%. From 2030, emissions in Poland should be reduced (compared to 2005) as follows: $SO_2$ by 70%, NOx by 39%, NMVOC by 26%, $NH_3$ by 15%, and PM2.5 by 58%. According to the National Energy and Climate Plan for 2021–2030, Poland has committed itself to reduce greenhouse gas emissions from non-ETS (ETS—EU Emission Trading Scheme) sectors by 7% compared to 2005 [18].

During the United Nations Conference on Climate Change, which took place in Paris in 2015, it was decided to take measures to prevent the global temperature rising by 2 °C compared to pre-industrial levels. Therefore, greenhouse gas emissions, including carbon dioxide, should be significantly reduced. The European Green Deal has provided a roadmap revealing intended actions to achieve its climate neutrality by 2050. The REFLEX Project (analysis of the European energy system under the aspects of flexibility and technological progress) was focused on the analysis and evaluation of the development towards a low carbon EU energy system. The project was carried out in 2016–2019 by nine universities and research institutes from Europe as part of the European Union's research and innovation program

"Horizon 2020". The results of the project included fuel consumption, technology development, and greenhouse gas (GHG) emission scenarios in various sectors by 2050 for EU-27 member states plus the United Kingdom, Norway, and Switzerland [19]. The proposed decarbonization scenarios were assessed using the comprehensive Energy Models System (EMS), which combined and coupled the single issue-specific models. The ASTRA (Assessment of Transport Strategies) model was dedicated to the transport sector, ELTRAMOD (Electricity Transshipment Model) to the electricity generation and transmission sectors, FORECAST to estimate future energy demand, and TIMES-HEAT-EU to centralized heat generation. A more detailed description of the models used in the REFLEX Project, model coupling, and data exchange (as well as the results of the REFLEX Project), taking changes in future fuel consumption in the EU into account, was given in [20]. The results of the REFLEX Project are also publicly available in the REFLEX Database [21].

The changes in the fuel and technologies structure of energy production have had an impact on the amount of emitted pollutants such as PM, $NO_x$, $SO_2$, and benzo(a)pyrene (B(a)P) that directly affect air quality and human health. The πESA platform (Platform for Integrated Energy System Analysis) developed at the AGH University of Science and Technology was used to study these effects [22]. The platform's concept is based on the Driver-Pressure-State-Impact-Response (DPSIR) framework. The DPSIR methodological framework is widely used to evaluate the impact of the developed energy scenarios on the inhabitant's health and external costs [23–25]. The DPSIR methodology was developed in the 1990s by the Organization for Economic Cooperation and Development (OECD). It has also been adopted by the EEA. In this work "Drivers" were represented by energy scenarios, and "Pressures" were linked to the emissions resulting from the scenarios developed in the REFLEX Project. "States" referred to the concentration and deposition of air pollutants that were calculated for Poland using the Polyphemus air quality modelling system [26–28]. "Impacts" considering health and external cost were estimated using the πESA platform. The data flow, used tools, and scope of work are presented in Figure 1. The calculation procedure is given in the methodology section.

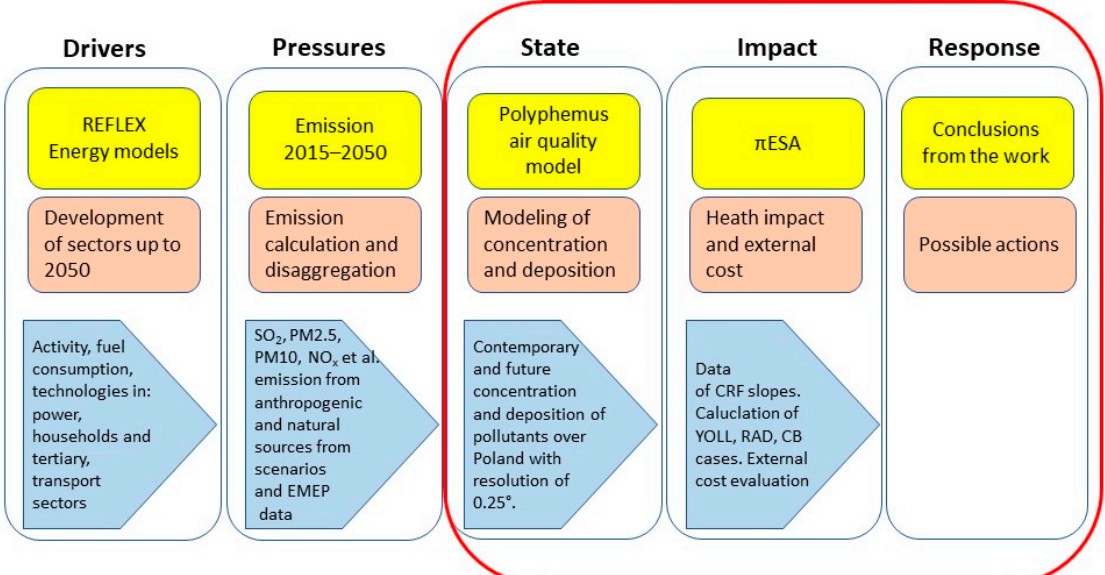

**Figure 1.** The Driver-Pressure-State-Impact-Response (DPSIR) framework, tools, data flow, and scope of the works. The red frame contains the elements that were made for the purpose of this work. EMEP: European Monitoring and Evaluation Programme; YOLL: Years of Life Lost; RAD: Restricted Activity Days; CB: Chronic Bronchitis; PM10: particulate matter with particles with a diameter of less than 10 μm; PM2.5: particulate matter with particles with a diameter of less than 2.5 μm; πESA: Platform for Integrated Energy System Analysis; CRF: concentration–response function.

From early 90s until 2005, a series of projects with a generic name "External Costs of Energy" (ExternE) aimed at developing the Impact-Pathway Approach were carried out. In general, the Impact-Pathway Approach is similar to the DPSIR framework employed in this work. In the frame of the ExternE projects, three models to calculate external cost were developed: EcoSense, EcoSenseLE, and RiskPoll. As the ExternE was very comprehensive and complex, the concentration–response functions developed therein are still widely used and were also applied in this work [29]. It is also worth mentioning that The International Institute for Applied Systems Analysis (IIASA) has been developing the GAINS (Greenhouse gas–Air pollution Interactions and Synergies) integrated assessment model since 2006. The GAINS model is a tool that allows one to assess the impact of climate policies on air quality and human health [30,31]. In the frame of the ENERGO (Earth Observation for monitoring and assessment of the environmental impact of energy use) Project, the impact of different emission scenarios up to 2050 on human health was analyzed [32,33]. Studies on human health and external costs of various pollutants have shown that the impact of PM2.5 is dominating [34]. All the aforementioned works employed a similar methodological approach that started with the determination of emission scenarios followed by air quality modelling and the assessment of health impacts due to population exposure to the elevated pollutant concentrations with the use of the dose–response functions.

The aim of this research was to investigate how fuel and technological transformations in the power, road transport, and household and tertiary sectors aimed to reduce $CO_2$ emissions in Poland would affect air quality, human health, and the associated external costs. Three energy scenarios developed within the REFLEX Project were analyzed. Emissions from the power, residential, household and tertiary, and industry sectors were taken into account. The impact of changes in PM2.5 concentration on human health and associated external costs was evaluated. The research was conducted for the year 2050, and the results were compared with the baseline year 2015.

## 2. Input Data: Energy and Emission Scenarios

In this work, the results of the energy and emission scenarios of the REFLEX Project were used as the input data. Energy and emission scenarios were developed using an innovative and comprehensive EMS developed within the REFLEX Project. The EMS combined and coupled the single issue-specific models from all the REFLEX partners. These models were dedicated to techno-economic learning, fundamental energy system modelling, and environmental impact assessment. To ensure a consistent system analysis, all models were based on a common database and scenario framework. Three scenarios of European decarbonization pathways were developed [19,20,35,36]. Firstly, the "moderate renewable scenario" (Mod-RES), representing development projection based on environmental and energy policies being decided or implemented at the time (year 2015). Secondly, the "high renewable scenario" (High-RES), which assumes new policies needed to limit the global warming to 2 °C by more rapidly reducing greenhouse gas emissions. This reduction was to be achieved by increasing share of renewable energy and flexible technologies to balance the intermittent character of renewable sources. Thirdly, the High-RES was split into two sub-scenarios, one called "High-RES centralized" in which energy production takes place predominantly in large centralized facilities and the other called "High-RES decentralized" in which energy production takes place more locally at the prosumers' installations. Table 1 presents the reductions of main pollutants in Poland in 2050 compared to 2005 for the aforementioned scenarios.

According to the results of the REFLEX Project simulation, the Polish power, household and tertiary sectors are no longer dependent on coal in 2050. The power sector is based on nuclear power, gas units, and renewable sources. In the residential and tertiary sectors, a drop in energy demand by about 40% between 2015 and 2050 was observed. Coal is being replaced by biomass and photovoltaics in these sectors. In the transport sector, BEV (Battery Electric Vehicle) and PHEV (Plug-in Hybrid Electric Vehicle) dominate the car fleet by 2050 [19,20,35,36].

**Table 1.** Reduction of pollutant emission in Poland in 2050 compared to 2005 taken into account in the analyses. Data for the moderate renewable scenario (Mod-RES) and the two high renewable scenarios (High-RES), decentralized and centralized, scenarios considered in the analyses (%). Based on [19,20,35,36].

| Pollutant | Sector | Mod-RES | High-RES Decentralized | High-RES Centralized |
|---|---|---|---|---|
| $CO_2$ | Power | 83 | 93 | 86 |
| | Household and tertiary | 53 | 97 | 98 |
| | Road transport | | 68 | |
| PM2.5 | Power | 66 | 82 | 98 |
| | Household and tertiary | 96 | 95 | 97 |
| | Road transport | | 52 | |
| $SO_2$ | Power | 99 | 99 | 99 |
| | Household and tertiary | 99 | 99 | 99 |
| $NO_X$ | Power | 66 | 86 | 79 |
| | Household and tertiary | 65 | 76 | 85 |
| | Road transport | | 68 | |

## 3. Methodology

### 3.1. Modeling of Air Quality

To model concentration and deposition of PM2.5 and other pollutants, the Polyphemus air quality modelling system was applied [26]. Polyphemus has been widely used by many research groups for many years, and its performance has been thoroughly evaluated. For instance, the evaluation of its results, including the concentration and deposition of PM (PM2.5 and PM10), $SO_2$, $NO_2$, $O_3$, and heavy metals against measurements of various types of stations (background, urban, industrial, and transport) can be found in [22,26,27,37].

The configuration of Polyphemus in this study included the Eulerian chemical-transport-model Polair3D, which was used for both gaseous and aerosol species. Polair3D tracked multiphase chemistry (gas, water, and aerosols). The model considered the following PM2.5 components: aromatics; primary and secondary organic aerosols; black carbon; mineral dust; $SO_4$, $NO_3$, and $NH_4$ aerosols; and hydrochloric acid aerosols. In the first step, Polyphemus was run over Europe, and the results from this simulation were used as the boundary conidiation data for a simulation over Poland. The European domain consisted of $34 \times 40$ cells starting from 35.9 °N latitude and 12 °W longitude, with a horizontal resolution of 1°. The domain over Poland had the geographical extent from 12.25° E–27.25° E of longitude to 46.75° N–56.75° N of latitude, with a horizontal resolution (along of longitude and latitude) of 0.25°. Five vertical levels were used with the following limits (in meters above surface): 0, 50, 600, 1200, 2000, and 3000. The results of simulation were saved each hour as the average value from six 10-min time steps. In total, twelve one-year simulations of the pollutant's dispersion were run, first with the use of the historical data reported to the European Monitoring and Evaluation Programme (EMEP) for 2015 for both domains [38]. Then, ten simulation were run over Poland. Three simulations were based on the emission data for the household and tertiary, power, and road transport sectors in 2050 for the Mod-RES, High-RES decentralized, and High-RES centralized scenarios. Next, six simulations were run separately for each sector (power and household and tertiary sectors) and scenario (Mod-RES, High-RES decentralized, and High-RES centralized). For the transport sector, only one simulation was run because emissions from this sector were the same for all scenarios. The national pollutant emissions calculated from the results of the REFLEX Project scenarios, which were provided for the power, household and tertiary, and road transport sectors, were horizontally gridded for each sector into cells with a resolution of 50 by 50 km based on the EMEP 2015 pattern [38]. This process was done by Polyphemus with the use of land use coverage (LUC) data with a horizontal resolution of 1 by 1 km [39]. The emissions from other sectors and natural emissions remained at the

2015 level. The meteorological conditions were calculated based on the data of European Centre for Medium-Range Weather Forecasts for 2008 [40]. These data were provided every 3 h with a horizontal resolution of 0.25° on 54 vertical levels.

### 3.2. Health Impacts and External Costs

Health impacts and external costs were calculated with the use of the πESA platform [22]. The approach employed in πESA is largely based on the methodology developed within a series of the ExternE projects [41,42]. Some of the most comprehensive and complex research in this area has been carried out within ExternE.

According to [6–9], fine particulate matter with an aerodynamic diameter of 2.5 mm or less (primary and airborne) is responsible for the most significant impacts to human health. Hence, the health impacts considered in this study were limited to people's long-term exposure to fine particulate (PM2.5) air pollution. The considered damages were expressed with the use of the standardized indicators (i) Years of Life Lost (YOLL), (ii) Chronic Bronchitis (CB), and (iii) Restricted Activity Days (RAD). YOLL is an estimate of the average years that a person would have lived if he or she had not died prematurely. CB refers to newly observed cases but not to a change in adult incidence rate. RAD corresponds to days when an individual's routine activities are disrupted, including days spent in bed, away from work or school, and days when activity is partially restricted due to sickness.

To link pollutant concentration with health impacts, concentration–response functions (CRFs) were used. CRFs relate the quantity of a pollutant that affects a population (accounting for the absorption of the pollutant from the air into the body) to the physical impact according to the following Equation (1):

$$I = Con \cdot Pop \cdot Fr \cdot CRF \tag{1}$$

where I is the health impact of a given type (i.e., YOLL, CB, and RAD) (cases); Con is the average annual concentration of PM2.5 at the ground level ($\mu g \cdot m^{-3}$), calculated with the use of Polyphemus air quality system; Pop is population exposed (number) based on [43,44]; Fr denotes the fraction of population affected (%), assumed to be 100%; and CRF is the concentration–response function for a given impact type ($cases.\mu g^{-1} \cdot m^3$), based on [45–47].

In order to calculate the overall external costs of air pollution, the calculated health impacts were assigned with the unit costs specific to each impact. In general, health costs are estimated by measuring the loss of direct peoples' income (e.g., lost wages or all medical treatment expenses over the patient's lifetime) or by capturing individuals' willingness to pay to avoid or reduce the risk of death or ill health [48]. Table 2 shows the values representing the CRFs' slopes and unit impact-specific costs used in this study. They were derived from the results of ExternE [45], NETCEN database [46], and the AQMEII3 initiative [47].

**Table 2.** Slopes and unit values of considered CRFs for PM2.5. Based on [45–47].

| CRF | CRF (Cases $\mu g \cdot m^{-3}$) | Unit Value (Eur2013case$^{-1}$ *) |
|---|---|---|
| PM2.5—Mortality YOLL | $3.42 \times 10^{-4}$ | 57,510 |
| PM2.5—Chronic Bronchitis | $3.90 \times 10^{-5}$ | 38,578 |
| PM2.5—Restricted activity days | $4.20 \times 10^{-2}$ | 98 |

\* Case means: YOLL, RAD, and CB.

In Equation (1), the factor that has the most impact on the results obtained is the population. In Poland, the average population density is 123 person/km$^2$. In cities, these values are much higher, e.g., it reaches 2384 person/km$^2$ in Kraków. Therefore, the same pollution concentration may cause several dozen times more health damage in cities than in rural areas. In the case of annual average concentrations of particulate matter, the differences over Poland are much lower and amount to about 8 $\mu g \cdot m^{-3}$. Therefore, it is important to achieve a pollution ambient concentration that is as low as

possible in an area with a high number of inhabitants (cities), because even a small difference in air quality can cause a significant reduction in number of observed negative human health effects. The CRF slopes for PM2.5, PM10, and TSP (total suspended particles) for various type of cases have been widely presented in literature, but the values have not differed significantly between particular studies.

The PM2.5 concentration values calculated with the Polyphemus system were combined, through the CRFs, with the population exposed to pollutants based on the population data for 2015 and 2050 [43,44]. The national population data were gridded into the Polyphemus cells based on the pattern presented in [44]. The resulting population density is presented in Figure 1. According to [43], the population of Poland will decrease from 38 million to approximately 34 million by 2050. The health impacts and external costs were calculated for all age-groups and grouped in fractions representing males and females aged 0–19 and 20–100+.

## 4. Results

The concentrations of PM2.5 in 2015 and 2050 in Poland and the impact of energy transformation in three sectors (power, road transport, and household and tertiary) for three scenarios (Mod-RES, High-RES decentralized, and High-RES centralized) considered in the REFLEX Project were modeled with the use of the Polyphemus air quality system. An example of the obtained results is presented in Figure 2. The ambient PM2.5 concentration at the surface level in 2015 based on the EMEP over Poland ($\mu g \cdot m^{-3}$) is shown in Figure 2a; the following Figure 2b–f shows the reduction of the PM2.5 concentration level compared to the one presented in Figure 2a, the higher the number points, the greater the reduction. At present, the most polluted area is the south region of Poland (Figure 2a). This region is characterized by a high population density and mountainous terrain. This means that significant emissions that occur there are not well-transported in the air and often accumulate once they are released. It was found that the PM2.5 concentration in this region will decrease by approximately 5 $\mu g \cdot m^{-3}$ in 2050 (Figure 2b) as a result of the implementation of analyzed scenarios. In relative terms, this region will experience the highest decline in PM2.5 concentration of approximately 25%, while the decrease will be approximately 20% in central Poland. The lowest reductions (around 5%) could be observed near the country borders (Figure 2f). The reduction will be particularly noticeable for the household and tertiary sector (Figure 2d). The observed changes by 2050 in the power sector were found to be insignificant (Figure 2c). This is because this sector, at present, needs to comply with strict emission limits and the PM2.5 emission reduction potential is rather limited compared to other sectors. The differences in the case of transport are slightly larger (Figure 2e). The modeled concentration due to the implementation of the Mod-RES, High-RES decentralized, and High-RES centralized scenarios were found to be similar (differences of around two percentage points between the scenarios).

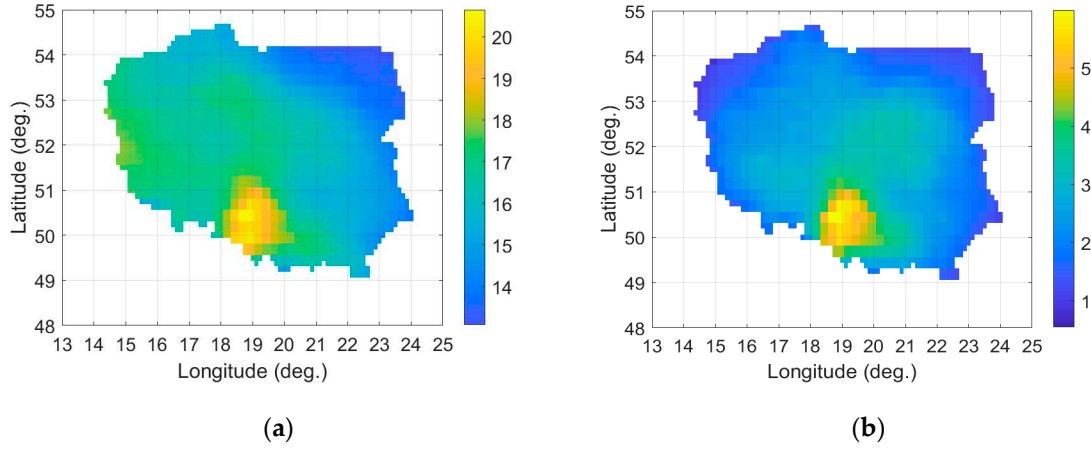

(a)　　　　　　　　　　　　　　　　　　　　　　　　　(b)

**Figure 2.** *Cont.*

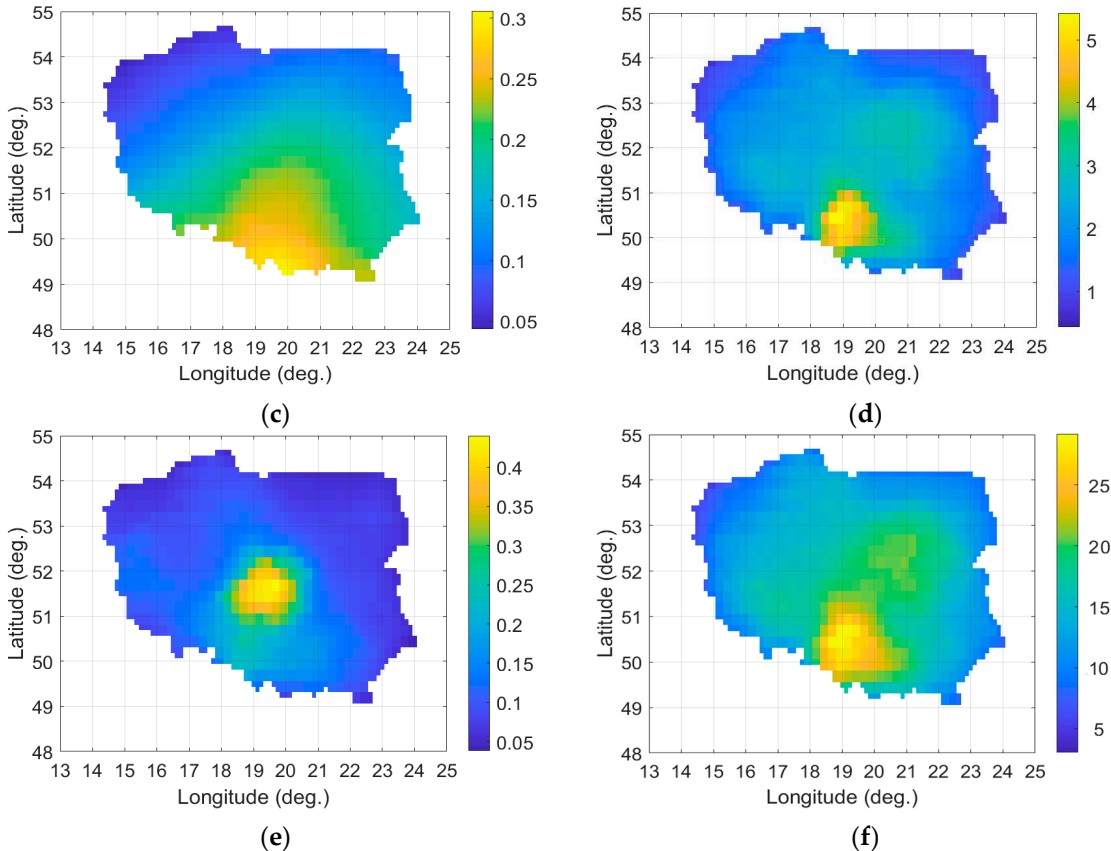

**Figure 2.** (**a**) The ambient PM2.5 concentration at surface level in 2015 based on the EMEP over Poland (μg·m$^{-3}$); (**b**) the difference between modelled ambient PM2.5 concentration at surface level in 2015 based on EMEP emission data and in 2050, where the emissions for the household and tertiary, road transport, and power sectors were applied according to the High-RES centralized scenario (μg·m$^{-3}$); (**c**) the difference between modelled ambient PM2.5 concentration at the surface level in 2015 based on EMEP emission data and in 2050, where the emissions for the power sector were applied according to the High-RES centralized scenario (μg·m$^{-3}$); (**d**) the difference between the modelled ambient PM2.5 concentration at the surface level in 2015 based on EMEP emission data and in 2050, where the emissions for the household and tertiary sector were applied according to the Mod-RES scenario (μg·m$^{-3}$); (**e**) the difference between the modelled ambient PM2.5 concentration at the surface level in 2015 based on EMEP emission data and in 2050, where the emissions for the road transport sector were applied according to the Mod-RES scenario (μg·m$^{-3}$); (**f**) the percentage change between the modelled ambient PM2.5 concentration at the surface level in 2015 based on EMEP emission data and in 2050, where the emissions for the household and tertiary, road transport, and power sectors were applied according to the High-RES centralized scenario (%). In Figure 2b–f, the higher the number points, the greater the reduction.

The total number of years of life lost due to the ambient PM2.5 concentration in Poland in 2015 was equal to 218,415 (Table 3). The number of restricted activity days equaled 73,487 years, and the number of new cases of chronic bronchitis reached the level of 24,907. The external cost of all these heath impacts was estimated at the level of 16.15 billion euros (the gross domestic product (GDP) of Poland in 2015 was EUR 430 billion). In the High-RES centralized scenario, the external costs were found to be reduced to 13.6 billion euros in 2050 (Table 4). The damage represented by the years of life lost had the highest share in external costs in both cases. Compared to 2015, the changes in the household and tertiary sector will lead to a reduction of more than 2.1 billion euro in all scenarios in 2050 (Table 5).

**Table 3.** The heath impact (YOLL—years of life lost; RAD—restricted activity days; and CB—chronic bronchitis) and external cost for each impact type in Poland associated with PM2.5 in 2015.

| Type of Data | Type of Impact | Units | Group | | | | Total |
| --- | --- | --- | --- | --- | --- | --- | --- |
| | | | Under 19 Years Old | | Over 20 Years Old | | |
| | | | Female | Male | Female | Male | |
| Health Impact | YOLL | years | 21,393 | 22,505 | 91,158 | 83,359 | 218,415 |
| | RAD | $10^3$ days | 2627 | 2764 | 11,194 | 10,237 | 26,823 |
| | CB | case | 2440 | 2566 | 10395 | 9506 | 24,907 |
| External Cost | YOLL | M€ | 1230 | 1294 | 5242 | 4794 | 12,561 |
| | RAD | M€ | 257 | 271 | 1097 | 1003 | 2629 |
| | CB | M€ | 94 | 99 | 401 | 367 | 961 |
| | Total | M€ | 1582 | 1664 | 6741 | 6164 | 16,150 |

**Table 4.** The heath impact (YOLL—years of life lost; RAD—restricted activity days; and CB—chronic bronchitis) and external cost for each impact type in Poland associated with PM2.5 in 2050, when the High-RES centralized scenario for the power, road transport, and household and tertiary sectors will be implemented.

| Type of Data | Type of Impact | Units | Group | | | | Total |
| --- | --- | --- | --- | --- | --- | --- | --- |
| | | | Under 19 Years Old | | Over 20 Years Old | | |
| | | | Female | Male | Female | Male | |
| Health Impact | YOLL | years | 18,532 | 19,483 | 76,510 | 69,732 | 184,257 |
| | RAD | $10^3$ days | 2276 | 2393 | 9396 | 8564 | 22,628 |
| | CV | case | 2113 | 2222 | 8725 | 7952 | 21,012 |
| External Cost | YOLL | M€ | 1066 | 1120 | 4400 | 4010 | 10,597 |
| | RAD | M€ | 223 | 234 | 921 | 839 | 2218 |
| | CB | M€ | 82 | 86 | 337 | 307 | 811 |
| | Total | M€ | 1370 | 1441 | 5658 | 5156 | 13,625 |

**Table 5.** The reduction of external cost between base year 2015 and 2050 according to the results of implementation of scenarios for considered sectors (M€).

| Scenario | Sector | Reduction of External Cost |
| --- | --- | --- |
| Mod-RES | Power | 145 |
| | Household and tertiary | 2146 |
| High-RES decentralized | Power | 144 |
| | Household and tertiary | 2124 |
| High-RES centralized | Power | 147 |
| | Household and tertiary | 2167 |
| | road transport | 122 |

## 5. Conclusions

In this study, the impacts of energy scenarios developed within the REFLEX Project up to 2050 for the power, road transport, and household and tertiary sectors on human health and associated external cost were analyzed. The analysis was performed in line with the DPSIR conceptual framework.

Poland faces a very big challenge in terms of reducing emissions of particulate matter, as well as other pollutants that directly affect air quality and greenhouse gas emissions. There are many scenarios, at both the national and international levels, that show how the energy transformation in Poland will progress in the future [49,50]. These scenarios most often focus on greenhouse gas emission reductions, as well as investment and operating costs, system flexibility, and energy security. However, it is also very important to consider how the solutions proposed in these scenarios contribute to the reduction of emissions of particulate matter and other pollutants affecting human health, as well as the associated

external costs. Due to the fact that, in most cases, GHG emitters also emit pollutants that have adverse impacts on air quality, the transition of energy systems towards the greater use of renewable energy sources will jointly contribute to climate change mitigation and cleaner air. This study shows that the greatest potential for reducing PM emissions in Poland is in the household and tertiary sector. The emission reduction potential is rather limited in the case of the power sector, which, at present, needs to comply with already-strict emission limits.

The results of this study show that despite the significant reduction of PM2.5 emissions, this reduction in emissions will lead to rather moderate changes in the average annual pollutant concentrations. The results show that the annual average PM2.5 ambient concentration over Poland will drop by a few $\mu g \cdot m^{-3}$ to a maximum of 5 $\mu g \cdot m^{-3}$. A significant impact of emission reductions can be observed in highly urbanized areas. This is due to the fact that, in this study, emissions from other sectors, natural emissions, and emissions in other countries were found to remain at the same levels in 2015 and 2050. On one hand, this shows that in order to improve air quality in Poland, measures must be taken in other sectors and neighboring countries. On the other hand, the most visible changes were still observed over Silesia and Lesser Poland, where air quality is the worst and mainly caused by emissions from the household and tertiary sector. As mentioned before, changes in these sectors will have a very positive impact on air quality in the future, not only in this region but also all over Poland.

The results of this study support the focus of the efforts that are currently being made to reduce PM emissions from the household and tertiary sector. The national plans envisage earmarking around EUR 25 billion of public funds for the renovation and replacement of heat sources in homes by 2030 [51]. The results indicate that by 2050, compared to 2005, the number of years of life lost will be reduced by 13% in case of people under 19 years of age and by 16% in case of people over 20 years of age. This will enable annual savings in external costs of EUR 2.4 billion in 2050 compared to 2015. This reduction in external costs would account for around 0.5% of Poland's current GDP. The results of the study show that external costs attributed to air pollution in 2015 were significant and exceeded 16 billion euros (more than 3% of Poland's GDP).

**Author Contributions:** Conceptualization, A.W. and J.Z.; methodology, J.Z., W.S., and A.W.; software, J.Z. and A.W.; validation, J.Z., T.O., and W.S.; formal analysis, M.P.; investigation, J.Z.; resources, W.S.; data curation, A.W., J.Z., and M.R.; writing—original draft preparation, J.Z., A.W., and M.R.; writing—review and editing, W.S., M.P., and T.O.; visualization, J.Z. and A.W.; supervision, W.S. and A.W. All authors have read and agreed to the published version of the manuscript.

**Funding:** This work received financial support from the funding of AGH University of Science and Technology, Faculty of Energy and Fuels (grant number 16.16.210.476). Part of the computational work was conducted using the resources of the Cyfronet AGH.

**Acknowledgments:** The authors would like to thank all colleagues from the REFLEX Project.

**Conflicts of Interest:** The authors declare no conflict of interest.

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
