# Peer review of "The Impact of Decarbonization Scenarios on Air Quality and Human Health in Poland—Analysis of Scenarios up to 2050"

_atmosphere, doi:10.3390/atmos11111222_

Round 1
Reviewer 1 Report
Dear Authors,
your paper is nice. Poor air quality in Poland is a really important issue not only for Polish people but also for neighboring countries. I have a few comments:
- Please write more about REFLEX project (1-2 sentences will be enough)
- Note the expansion to the abbreviations. They are often missing. This also applies to chemical compounds.
- (42) It would be good to mention the source of this information.
- (115, 116) I don't understand this sentence. Please, rewrite it.
- (134) You missed a closing punctuation.
- Please, check your citations. There is a lot of „Error! Reference source not found” in this text.
- Table 1 - What about SO2 pollutions from road transport? Is there no data available?
- (238, 239) Please, rewrite this sentence.
- Also pay attention to the way the data is presented in the tables. Large numbers can be difficult to read.
Wish you all the best
Author Response
Dear Authors,
your paper is nice. Poor air quality in Poland is a really important issue not only for Polish people but also for neighboring countries. I have a few comments:
- Please write more about REFLEX project (1-2 sentences will be enough)
- Note the expansion to the abbreviations. They are often missing. This also applies to chemical compounds.
- (42) It would be good to mention the source of this information.
- (115, 116) I don't understand this sentence. Please, rewrite it.
- (134) You missed a closing punctuation.
- Please, check your citations. There is a lot of „Error! Reference source not found” in this text.
- Table 1 - What about SO2 pollutions from road transport? Is there no data available?
- (238, 239) Please, rewrite this sentence.
- Also pay attention to the way the data is presented in the tables. Large numbers can be difficult to read.
Wish you all the best
Thank you!
- Please write more about REFLEX project (1-2 sentences will be enough)
The sentence has been added.
The project was carried out in 2016-2019 by nine universities and research institutes from Europe as part of the European Union’s research and innovation programme “Horizon 2020”.
- Note the expansion to the abbreviations. They are often missing. This also applies to chemical compounds.
The text has been corrected.
- (42) It would be good to mention the source of this information.
The refences have been added:
Chief Inspectorate of Environmental Protection Air quality assessment in zones in Poland for 2019 (in Polish).; Warszawa, 2020;
Bebkiewicz, K.; Dębski, B.; Chłopek, Z.; Doberska, A.; Kanafa, M.; Kargulewicz, I.; Olecka, A.; Rutkowski, J.; Skośkiewicz, J.; Waśniewska, S.; et al. Poland`s Informative Inventory Report 2019. Submission under the UN ECE Convention on Long-range Transboundary Air Pollution and the Directive (EU) 2016/2284; National Centre for Emissions Management (KOBIZE): Warsaw, 2019;
- (115, 116) I don't understand this sentence. Please, rewrite it.
The original sentence upload to the publisher was “From early 90s till 2005 a several projects called ExternE External Costs of Energy” aimed at developing the Impact-Pathway-Approach have been carried out. Impact-Pathway-Approach is similar to DPSIR framework employed in this work.”. In the publisher’s word processing system, the beginning of the sentence “From early 90s till 2005” was unfortunately automatically changed to “Figure 90. s till 2005”. In pdf version uploaded to the system this error did not appear.
- (134) You missed a closing punctuation.
Thank you. The text has been corrected.
- Please, check your citations. There is a lot of „Error! Reference source not found” in this text.
Thank you. The problem seems to appear only in the Word version; the pdf uploaded to the publisher system does not have these errors. We have checked and corrected citations.
If similar problems occur again, we will work together with the Atmosphere editorial team to resolve such errors.
- Table 1 - What about SO2 pollutions from road transport? Is there no data available?
The SO2 from transport was not considered – the emission is very low and SO2 emission standards are not set for this sector
- (238, 239) Please, rewrite this sentence.
The sentence has been changed,
- Also pay attention to the way the data is presented in the tables. Large numbers can be difficult to read.
Tables 3 and 4 have been corrected.

Reviewer 2 Report
In my opinion, the manuscript submitted for review is well prepared and suitable for publication in the Atmosphere. The contents of the manuscript are interesting and should be printed.
The only item for improvement is the line 115, where is: Figure 90. s till 2005 a several projects called ExternE “External Costs of Energy” aimed at developing the ......; should be corrected, because Figure 90 does not exist in the text. I think should be “In years ....
The authors describe the purpose in detail in their work. They accurately present the models on the basis of which they make calculations and forecasts. The tables contain well-described data.
The individual stages of work and the course of the forecasting procedure are very well presented.
The mathematical formulas and how to use them are explained in detail.
The authors pointed out that the reduction of emissions from the household sector will have the greatest impact on the improvement of air quality and life expectancy of people.
The article is interesting, the presented scenarios are credible.
The authors note that after reducing the concentration of PM2.5 dust, the total reduction of dust in the atmosphere will not bring significant improvement.
This is due to both the emission of natural dust into the air and the emission of dust in countries neighboring Poland.
The publication of this article should contribute to greater cooperation between countries to undertake joint actions to improve air quality.

Author Response
Comments and Suggestions for Authors
In my opinion, the manuscript submitted for review is well prepared and suitable for publication in the Atmosphere. The contents of the manuscript are interesting and should be printed.
Dear Reviver. Thank you.
The only item for improvement is the line 115, where is: Figure 90. s till 2005 a several projects called ExternE “External Costs of Energy” aimed at developing the ......; should be corrected, because Figure 90 does not exist in the text. I think should be “In years ....
The original sentence upload to the publisher was “From early 90s till 2005 a several projects called ExternE External Costs of Energy” aimed at developing the Impact-Pathway-Approach have been carried out. Impact-Pathway-Approach is similar to DPSIR framework employed in this work.”. In the publisher’s word processing system, the beginning of the sentence “From early 90s till 2005” was unfortunately automatically changed to “Figure 90. s till 2005”. In pdf version uploaded to the system this error did not appear. If similar problems occur again, we will work together with the Atmosphere editorial team to resolve such errors.
The authors describe the purpose in detail in their work. They accurately present the models on the basis of which they make calculations and forecasts. The tables contain well-described data.
The individual stages of work and the course of the forecasting procedure are very well presented.
The mathematical formulas and how to use them are explained in detail.
The authors pointed out that the reduction of emissions from the household sector will have the greatest impact on the improvement of air quality and life expectancy of people.
The article is interesting, the presented scenarios are credible.
The authors note that after reducing the concentration of PM2.5 dust, the total reduction of dust in the atmosphere will not bring significant improvement.
This is due to both the emission of natural dust into the air and the emission of dust in countries neighboring Poland.
The publication of this article should contribute to greater cooperation between countries to undertake joint actions to improve air quality.

Reviewer 3 Report
The aritcle: The Impact of Decarbonization Scenarios on Air Quality and Human Health in Poland - Analysis of Scenarios up to 2050
The introduction lacks enough citations for some very important claims that the authors make.
English writing needs to be improved throughout the manuscript.
The introductory section is not easy to read and needs better connectors between ideas. It needs to improve its organization. The information presented does not flow naturally into the aim of the project.
A good discussion section (that can be part of the results as well) is missing from this manuscript. The authors show a good amount of numbers and results that sometimes are not well connected to a description of the local context. Additionally, there are no references to how other countries have calculated the same (or similar) effects.
some other comments:
Line 19: please define the REFLEX project
Line 27: briefly explain what the polyphemus air quality sistem is
Line 28: the same for pi-ESA platform
Line 39: please provide some information that supports this claim. Most polluted in terms of what? compared to? how much greater?
Line 40: define CAFE
line 41: what is this limit?
Line 42: include a citation to support that claim
Line 57: a citation is needed
Line 67: a citation is needed
Line 115: parts of the text appear to be missing
Line 151, 245, 247, 254, 255, 259, 262, etc: error in the references
Figure 2: it is hard to interpret the differences between 2015 and 2050. The higher the number the larger the reduction? I suggest clarifying this, particularly the color scales.
Author Response
The aritcle: The Impact of Decarbonization Scenarios on Air Quality and Human Health in Poland - Analysis of Scenarios up to 2050
The introduction lacks enough citations for some very important claims that the authors make.
Thank you.
The following citations have been added.
Chief Inspectorate of Environmental Protection Air quality assessment in zones in Poland for 2019 (in Polish).; Warszawa, 2020;
Nakane, H. Translocation of particles deposited in the respiratory system: a systematic review and statistical analysis. Environ Health Prev Med. 2012, 17(4), 263–274.
Crippa, M.; Janssens-Maenhout, G.; Guizzardi, D.; Dingenen, R.V.; Dentener, F. Contribution and uncertainty of sectorial and regional emissions to regional and global PM2:5 health impacts. Atmos. Chem. Phys. 2019, 5165–5186.
PSE-Operator Polish Power System - Report 2019. Summary of quantitative data on functioning of Polish Power System in 2019 (in Polish).; Warszawa, 2020;
English writing needs to be improved throughout the manuscript.
Thank you. The text has been reviewed in terms of grammar and structure.
The introductory section is not easy to read and needs better connectors between ideas. It needs to improve its organization. The information presented does not flow naturally into the aim of the project.
Thank you. The text has been changed. We have tried to clarify some abbreviations. We tried to explain our way of thinking as best as possible, so we included figure 1. The work combines many aspects, we tried to mention all of them.
A good discussion section (that can be part of the results as well) is missing from this manuscript. The authors show a good amount of numbers and results that sometimes are not well connected to a description of the local context. Additionally, there are no references to how other countries have calculated the same (or similar) effects.
The text has been corrected and some sentences have been added.
We do not directly compare our results with other studies, because the complexity of each project, the differences in assumptions and policy scenarios make such comparisons unsubstantiated without a thorough and detailed analysis of all relevant factors, which we believe is the topic for a separate review article.
some other comments:
Line 19: please define the REFLEX project
The full name of the REFLEX project has been added. (Analysis of the European energy system under the aspects of flexibility and technological progress).
Additionally, in introduction the sentence has been added.
“The project was carried out in 2016-2019 by nine universities and research institutes from Europe in frame of the European Union’s Horizon 2020 research and innovation programme.”
Line 27: briefly explain what the polyphemus air quality sistem is
One sentence has been changed. “Next, Based on the for each emission scenarios, ambient concentrations of PM2.5 and others pollutants were modelled calculated with the use of the Polyphemus – an Eulerian-type air quality modelling system.”
Line 28: the same for pi-ESA platform
. The full name of the pi-ESA platform has been added.
“Platform for Integrated Energy System Analysis”
Line 39: please provide some information that supports this claim. Most polluted in terms of what? compared to? how much greater?
Line 40: define CAFÉ
The text has been corrected.
The limit values set for the ambient concentration of pollutants considered in the directives on ambient air quality and cleaner air for Europe (CAFE) are often exceeded
line 41: what is this limit?
The text has been corrected.
The limit values set for the ambient concentration of pollutants considered in the directives on ambient air quality and cleaner air for Europe (CAFE) are often exceeded
Line 42: include a citation to support that claim
The following citations have been added.
EEA Air quality in Europe - 2019 report. EEA Report No 10/2019. 2019
Chief Inspectorate of Environmental Protection Air quality assessment in zones in Poland for 2019 (in Polish).; Warszawa, 2020;
Line 57: a citation is needed
The following citations have been added.
Nakane, H. Translocation of particles deposited in the respiratory system: a systematic review and statistical analysis. Environ Health Prev Med. 2012, 17(4), 263–274.
Crippa, M.; Janssens-Maenhout, G.; Guizzardi, D.; Dingenen, R.V.; Dentener, F. Contribution and uncertainty of sectorial and regional emissions to regional and global PM2:5 health impacts. Atmos. Chem. Phys. 2019, 5165–5186.
Line 67: a citation is needed
The following citation has been added.
PSE-Operator Polish Power System - Report 2019. Summary of quantitative data on functioning of Polish Power System in 2019 (in Polish).; Warszawa, 2020;
Line 115: parts of the text appear to be missing
The original sentence upload to the publisher was “From early 90s till 2005 a several projects called ExternE External Costs of Energy” aimed at developing the Impact-Pathway-Approach have been carried out. Impact-Pathway-Approach is similar to DPSIR framework employed in this work.”. In the publisher’s word processing system, the beginning of the sentence “From early 90s till 2005” was unfortunately automatically changed to “Figure 90. s till 2005”. In pdf version uploaded to the system this error did not appear.
Line 151, 245, 247, 254, 255, 259, 262, etc: error in the references.
Thank you. The problem seems to appear only in the Word version; the pdf uploaded to the publisher system does not have these errors. We have checked and corrected citations.
If similar problems occur again, we will work together with the Atmosphere editorial team to resolve such errors.
Figure 2: it is hard to interpret the differences between 2015 and 2050. The higher the number the larger the reduction? I suggest clarifying this, particularly the color scales.
Thank you. We considered many different ways of presenting the results in graphic form, and we believe that the current form is adequate.
Following you suggestion the clarifying sentence “In the figures b,c,d,e and f the higher the number points, the greater the reduction“ has been added.

Reviewer 4 Report
Accept the paper.
Author Response
Dear Reviewer.
Thank you very much, Wish you all the best
This manuscript is a resubmission of an earlier submission. The following is a list of the peer review reports and author responses from that submission.
Round 1
Reviewer 1 Report
The manuscript “The Impact of CO2 Reduction on Air Quality and Human Health in Poland - Analysis of Scenarios up to 2050” present a very interesting topic which can help estimate the reduction of external cost from reduced pollutant emissions. However, the materials needs to be re-ordered significantly before it is ready for publication as it is not written in a scientific way, such as its missing context, lack of clarity on analysis and poor presentation style. Also the figures quality needs to be improved as they are poorly presented. The most serious problem to my opinion is that the title, background, results and conclusions of the manuscript may not to be convincing as the results seem not serve for the aims and unclear presented. The aim and results have to be correlated each other.The title is very missing leading. From the title I expected the main body of work is about greenhouse gas, while the results only presents the PM but the CO2 is missing. What is the πESA platform? Is the applied model possible to calculate the CO2? This work uses a lots of data from previous studies, but the references were not cited clear.There are a lots of type setting problems in the paper. The equation 1 is the main work in this study, however all the input data comes from other studies. In the methodology, why the male and female are distingushed, what is the difference? Also it will be necessary to discuss the influence of each factors to the health impact. Figure 2 should add the units the legneds. The results in the tables only shows 2015 and 2050. it would be interesting to show the the annual changes from 2015 to 2050 using the curves instead of tables. The reference formats should be uniformed.Author Response
Dear Reviewer.
Thank you for your time and valuable comments. We have adjusted the text according to your remarks.
The manuscript “The Impact of CO2 Reduction on Air Quality and Human Health in Poland - Analysis of Scenarios up to 2050” present a very interesting topic which can help estimate the reduction of external cost from reduced pollutant emissions. However, the materials needs to be re-ordered significantly before it is ready for publication as it is not written in a scientific way, such as its missing context, lack of clarity on analysis and poor presentation style. Also the figures quality needs to be improved as they are poorly presented.
We agree with the Reviewer remarks. The text has been revised. The additional figures have been added.
The most serious problem to my opinion is that the title, background, results and conclusions of the manuscript may not to be convincing as the results seem not serve for the aims and unclear presented. The aim and results have to be correlated each other.The title is very missing leading. From the title I expected the main body of work is about greenhouse gas, while the results only presents the PM but the CO2 is missing.
The Reviewer is right about misleading title. The title has been changed to better represent the content of the manuscript. The discussion of the results has been changed.
We would like to specify, that the CO2 concentration is not modelled in our work. The Polyphemus air quality system was used to model PM2.5 concentration in regional scale.
What is the πESA platform? Is the applied model possible to calculate the CO2?
More detailed description of πESA platform has been added.
The πESA is platform created to calculate the human health effects and external cost. The platform is being developed by authors since 2014. This platform was developed as the part of the system to cover the Drivers-Pressures-State-Impact-Response (DPSIR) framework. The system consists 3 main modules: (i) TIMES-PL (energy model developed by authors based on TIMES model generator), to create energy and emission scenarios, (ii) Polyphemus air quality system (developed in France in CEREA lab, at AGH adapted to Poland) to calculate concentration and deposition of pollutants such as PM, SO2, NOX, O2, heavy metals and (iii) MAEH - model for assessing the impact of pollutant emissions on the environment and human health. Due to the fact that CO2 does not directly affect human health and is a global pollutant, it is not possible to calculate the impact of CO2 by the πESA platform. Additionally, it is not possible to calculate CO2 concentrations over Poland with the use of Polyphemus, for this a global system would be needed (in the case of CO2, there are no large local deviations from the average concentration in the atmosphere).
This work uses a lots of data from previous studies, but the references were not cited clear.There are a lots of type setting problems in the paper. The equation 1 is the main work in this study, however all the input data comes from other studies.
In equation 1, the refence has been added. For the equation 1 the concentration (CON) were calculated in this work.
In the methodology, why the male and female are distingushed, what is the difference?
There is no difference in methodology of calculation health impact for male and female. The results are divided into male and female, since the input data on population density was divided into male and female.
Also it will be necessary to discuss the influence of each factors to the health impact. Figure 2 should add the units the legneds.
We agree with the Reviewer. The discussion of the results has been enhanced by the influence of each factor.
The results in the tables only shows 2015 and 2050. it would be interesting to show the the annual changes from 2015 to 2050 using the curves instead of tables.
Thank you for this remark. The simulations with the use of Polyphemus air quality system were done only for 2015 and 2050 for computational reasons. For this two datayears only 21 simulations were completed. As one simulation run takes around 1 week with the use of supercomputer (CYFRONET AGH), it is not possible for us of to run simulations for each year.
The reference formats should be uniformed.
We agree with the Reviewer. The reference formatting has been unified.

Reviewer 2 Report
There are several higher-level issues with the paper in its current format.
- How does the manuscript advance or contribute to the current state of science on sustainability? Literature review for future scenario modeling is almost non-existent in this paper. Why did the authors not include any relevant studies that conducted such future-scenario air quality simulations? It is very important to compare and contrast this study with earlier and related studies.
- The author's seem to have relied on previous or parallel studies to describe their study methodology and results. As such there is not a great of stand-alone contribution of this manuscript. The author's need to provide a lot more detail to the methods section and provide information on models used in this study.
- How did you validate the air pollution dispersion model? Since the base year is 2015 you will have the data to validate the model.
- The manuscript is replete with grammatical, spelling, and formatting issues. The manuscript needs to be thoroughly proof-read and corrected for these issues.
In addition to the above, below are a few lower-level details that I came across while reviewing:
Abstract: It will help readers if authors can provide higher level information on what the defining characteristics of the scenarios are in the abstract.
Line 25: "…transformation of households and tertiary sectors" is very vague. What does it mean?
Line 40&43: I think this is carcinogenic instead of carcinogenicity.
Line 42: "Similar shares of these sectors are recorded…" does not make clear sense. Please rephrase.
Line 49: Lungs instead of lugs
Lines 73-85: This paragraph is difficult to understand. Please think of readers who are not familiar with the REFLEX system. I think it makes sense to give overarching information about REFLEX plus the sub-models under it.
Lines 99-102: I don't understand what the authors are trying to say here: "The aim of this work was to investigate how transformation of the power, households, tertiary and road transport sectors heading towards the reduction of greenhouse gas emissions developed in the REFLEX Project affects concentration of PM2.5 in Poland."
Line 105: emission
Lines 106 and 112: Use of word elaborated seems awkward. Perhaps developed is a better alternative?
Line 120: Reference error?
Lines 123-127: I don't understand this paragraph. How do the authors claim the reduction in emissions shown in Table 1? Are these the results of a different study? If so they need to cite it but more importantly put the current study in the context of the older study.
Lines 152-154: It is difficult to follow the writing in these lines. It looks like for road transport sector the authors conducted only mod-res simulation. Why were the other two scenarios not run for road transport sector?
Fig 2d and 2e: The caption says these are differences in emissions between 2015 and 2050 for High-Res scenario. Is this centralized or decentralized?
Author Response
Dear Reviewer.
Thank you for your time and valuable comments. We have adjusted the text according to your remarks.
There are several higher-level issues with the paper in its current format.
- How does the manuscript advance or contribute to the current state of science on sustainability? Literature review for future scenario modeling is almost non-existent in this paper. Why did the authors not include any relevant studies that conducted such future-scenario air quality simulations? It is very important to compare and contrast this study with earlier and related studies.
Thank you for this remark. The review of similar projects regarding air quality simulations has been added. We do not directly compare our results with other studies, because the complexity of each project, the differences in assumptions and policy scenarios make such comparisons unsubstantiated without a thorough and detailed analysis of all relevant factors, which we believe is the topic for a separate review article.
- The author's seem to have relied on previous or parallel studies to describe their study methodology and results. As such there is not a great of stand-alone contribution of this manuscript. The author's need to provide a lot more detail to the methods section and provide information on models used in this study.
The changes have been made to indicate the scope of this work. The work uses the previous works of the authors who have been developing individual analyses and modelling tools for over a dozen years, but many works was done specifically for this article.
- How did you validate the air pollution dispersion model? Since the base year is 2015 you will have the data to validate the model.
The Polyphemus air quality model has been validated in detail over Poland and Europe. In this work we took the metrological data for 2010 (as the repressive year for our study) and emission from 2015, so we do not validate this data against measurements. The validation of system can be found in previous works of authors of this article and other authors. For example:
- Zyśk, J.; Stężąły, A.; Pluta, M.; Wyrwa, A.; Roustan, Y.; Sportisse, B. The Polyphemus system to model of transport of pollutants (in Polish).; Aktualne problemy w ochronie powietrza atmosferycznego.; Polskie Zrzeszenie Inżynierów i Techników Sanitarnych.: Wrocław, 2008;
- Wyrwa, A. An optimization platform for Poland’s power sector considering air pollution and health effects. Environmental Modelling & Software 2015, 74, 227–237.
- Zyśk. PhD thesis (AGH and Pairs-Est). “Modelling of atmospheric transport of heavy metals emitted from Polish power sector”
- Zyśk, J.; Roustan, Y.; Wyrwa, A. Modelling of the atmospheric dispersion of mercury emitted from the power sector in Poland. Atmospheric Environment 2015, 112, 246–256.
- The manuscript is replete with grammatical, spelling, and formatting issues. The manuscript needs to be thoroughly proof-read and corrected for these issues.
Thank you. The text has been corrected.
In addition to the above, below are a few lower-level details that I came across while reviewing:
Abstract: It will help readers if authors can provide higher level information on what the defining characteristics of the scenarios are in the abstract.
Thank you. The text has been corrected.
Line 25: "…transformation of households and tertiary sectors" is very vague. What does it mean?
Thank you. The text been corrected.
Line 40&43: I think this is carcinogenic instead of carcinogenicity.
Thank you. The text been corrected.
Line 42: "Similar shares of these sectors are recorded…" does not make clear sense. Please rephrase.
Thank you. The text has been corrected. We had twice this sentence (one by mistake).
Line 49: Lungs instead of lugs
Thank you. The text has been corrected.
Lines 73-85: This paragraph is difficult to understand. Please think of readers who are not familiar with the REFLEX system. I think it makes sense to give overarching information about REFLEX plus the sub-models under it.
Thank you. The text has been corrected.
Lines 99-102: I don't understand what the authors are trying to say here: "The aim of this work was to investigate how transformation of the power, households, tertiary and road transport sectors heading towards the reduction of greenhouse gas emissions developed in the REFLEX Project affects concentration of PM2.5 in Poland."
Thank you. This paragraph has been rephrased.
Line 105: emission
Thank you. The text has been corrected.
Lines 106 and 112: Use of word elaborated seems awkward. Perhaps developed is a better alternative?
Thank you. The text has been corrected.
Line 120: Reference error?
Thank you. The text has been corrected.
Lines 123-127: I don't understand this paragraph. How do the authors claim the reduction in emissions shown in Table 1? Are these the results of a different study? If so they need to cite it but more importantly put the current study in the context of the older study.
The structure of article has been changed. The refence has been added.
Lines 152-154: It is difficult to follow the writing in these lines. It looks like for road transport sector the authors conducted only mod-res simulation. Why were the other two scenarios not run for road transport sector?
The text has been corrected. Only one scenario was developed for transport. Mod-RES, High-RES decentralized, High-RES centralized differ in the use of various fuels and technologies in power, household and tertiary sectors. There is no difference in transport sector between scenarios.
Fig 2d and 2e: The caption says these are differences in emissions between 2015 and 2050 for High-Res scenario. Is this centralized or decentralized?
Thank you. The text has been corrected. It is Mod-RES scenario.
